# A mitofusin-dependent docking ring complex triggers mitochondrial fusion *in vitro*

Tobias Brandt[1†], Laetitia Cavellini[2†], Werner Kühlbrandt[1*], Mickaël M Cohen[2*]

[1]Max Planck Institute of Biophysics, Frankfurt, Germany; [2]Laboratoire de Biologie Moléculaire et Cellulaire des Eucaryotes, Institut de Biologie Physico-Chimique, Sorbonne Universités, Paris, France

**Abstract** Fusion of mitochondrial outer membranes is crucial for proper organelle function and involves large GTPases called mitofusins. The discrete steps that allow mitochondria to attach to one another and merge their outer membranes are unknown. By combining an *in vitro* mitochondrial fusion assay with electron cryo-tomography (cryo-ET), we visualize the junction between attached mitochondria isolated from *Saccharomyces cerevisiae* and observe complexes that mediate this attachment. We find that cycles of GTP hydrolysis induce progressive formation of a docking ring structure around extended areas of contact. Further GTP hydrolysis triggers local outer membrane fusion at the periphery of the contact region. These findings unravel key features of mitofusin-dependent fusion of outer membranes and constitute an important advance in our understanding of how mitochondria connect and merge.

**\*For correspondence:** werner.
kuehlbrandt@biophys.mpg.de
(WK); cohen@ibpc.fr (MMC)

[†]These authors contributed
equally to this work

**Competing interest:** See
page 21

**Reviewing editor:** Nikolaus
Pfanner, University of Freiburg,
Germany

## Introduction

Membrane fusion underlies fundamental biological processes such as fertilization, virus entry into host cells or intracellular protein trafficking. Protein and lipid trafficking mainly involves SNAREs (Soluble N-ethyl maleimide sensitive factor Attachment protein Receptors) that are expressed on all intracellular compartments undergoing fusion except peroxisomes and mitochondria (*Cai et al., 2007*; *Escobar-Henriques and Anton, 2013*).

Mitochondria constitute a remarkably dynamic network with an organization and ultra-structure that is regulated by fusion and fission of mitochondrial outer and inner membranes (*Labbé et al., 2014*; *Westermann, 2010*). Fusion and fission are crucial for all mitochondrial functions including oxidative phosphorylation, calcium signalling, apoptosis and lipid metabolism. Defects in mitochondrial fusion and fission are associated with numerous pathologies and severe neurodegenerative diseases (*Dorn, 2013*; *Liesa et al., 2009*).

Mitochondrial fusion and fission both depend on large GTPases of the Dynamin-Related Protein (DRP) family (*Labbé et al., 2014*; *Low and Löwe, 2010*). To promote fission, soluble DRPs assemble into spirals around membrane compartments. GTP hydrolysis causes the spirals to constrict, reducing the diameter of the compartments, ultimately followed by their separation (*Bui and Shaw, 2013*). In contrast to fission, the role of DRPs in lipid bilayer fusion remains poorly understood. Among the three families of transmembrane DRPs implicated in fusion, Mitofusins and OPA1 mediate fusion of the mitochondrial outer and inner membranes, respectively, whereas atlastins promote homotypic fusion of ER tubules (*McNew et al., 2013*). GTP binding and hydrolysis participate in *trans* auto-oligomerization of atlastins and OPA1 through their respective GTPase domains (*Rujiviphat et al., 2012*; *Klemm et al., 2011*; *Byrnes et al., 2013*; *DeVay et al., 2009*; *Meglei and McQuibban, 2009*; *Moss et al., 2011*; *Saini et al., 2014*). The resulting homotypic tethering of ER

**eLife digest** Yeast and other eukaryotic cells contain distinct compartments that have specific roles. For example, compartments called mitochondria – which are surrounded by two layers of membrane – provide the energy needed for many cell processes. The organization of the network of mitochondria in a cell has a large effect on their capacity to provide energy. Mitochondria can fuse together to make larger compartments or divide to make smaller ones. Defects in fusion or division of mitochondria can reduce the amount of energy that is provided, which, in humans and animals can lead to diseases that affect various organs, especially those in the nervous system.

When two mitochondria fuse they must first attach to each other and then merge their outer membranes. Proteins called mitofusins are known to be involved in these processes, but the molecular details of how they take place were not clear.

Brandt, Cavellini *et al.* investigated how mitochondria isolated from budding yeast cells attach to each other. The experiments found that two mitochondria first become loosely attached by mitofusins. These proteins then promote a tighter attachment in which the outer membranes of the two mitochondria come into contact over a larger area. This contact area is determined by a linear arrangement of proteins referred to as the docking ring. Brandt, Cavellini *et al.* further observed that local fusion between the outer membranes takes place at the edge of the contact area in the path of the docking ring.

Future research will need to address how mitochondria attach to each other in living cells and how the process is regulated.

and mitochondrial inner membranes is accompanied by conformational rearrangements of the DRPs that are thought to trigger subsequent fusion of lipid bilayers. Based on structural insights gained from BDLP (Bacterial Dynamin-Like Protein) (*Low and Löwe, 2006*; *Low et al., 2009*), a close relative of the yeast mitofusin Fzo1, mitofusins may promote outer membrane tethering and fusion through similar processes of oligomerization and conformational rearrangement (*Cohen et al., 2011*; *Escobar-Henriques and Anton, 2013*).

As seen with spirals formed during membrane scission, DRPs are characterized by their ability to assemble into higher-order macromolecular structures (*Ingerman et al., 2005*; *Low et al., 2009*; *Mears and Hinshaw, 2008*). Whether DRPs participate in the formation of such structures during membrane attachment and fusion is unknown. In particular, the precise orchestration of events from the initial attachment of membranes to their ultimate fusion and the requirement for GTP binding and hydrolysis during these steps is elusive. By combining the power of an *in vitro* mitochondrial fusion assay with cryo-electron tomography (cryo-ET), we undertook to visualize the junction between mitochondria, resulting in an unprecedented dissection of the outer membrane fusion process.

## Results

### Cryo-ET reveals distinct populations of attached mitochondria

The *in vitro* mitochondrial fusion assay (*Figure 1A*) allows us to distinguish between discrete steps of the mitochondrial fusion process (*i.e.* attachment, fusion of outer membranes and fusion of inner membranes) (*Hoppins et al., 2009*; *Meeusen et al., 2006*, *2004*). Purified mitochondria from wild-type yeast cells were brought into contact by centrifugation and were incubated at 4°C for 10 min to promote mitofusin-dependent attachment (*Meeusen et al., 2004*). Subsequent incubation for 45 min at 25°C allows fusion of outer membranes but not inner membranes (*Figure 1A*; top) unless energy is regenerated (*Meeusen et al., 2004*). Consistent with this, fusion reactions recurrently yielded 6 to 8% intermediates with fused outer membranes (*Figure 1B*). Prolonged incubation of centrifuged mitochondria at 4°C (*Figure 1A*; bottom) decreases fusion of outer membranes but stabilizes attached intermediates (*Cohen et al., 2011*) with approximately 25% of mitochondria in close contact (*Figure 1C*). We reasoned that with an imaging technique of sufficient resolution, it should

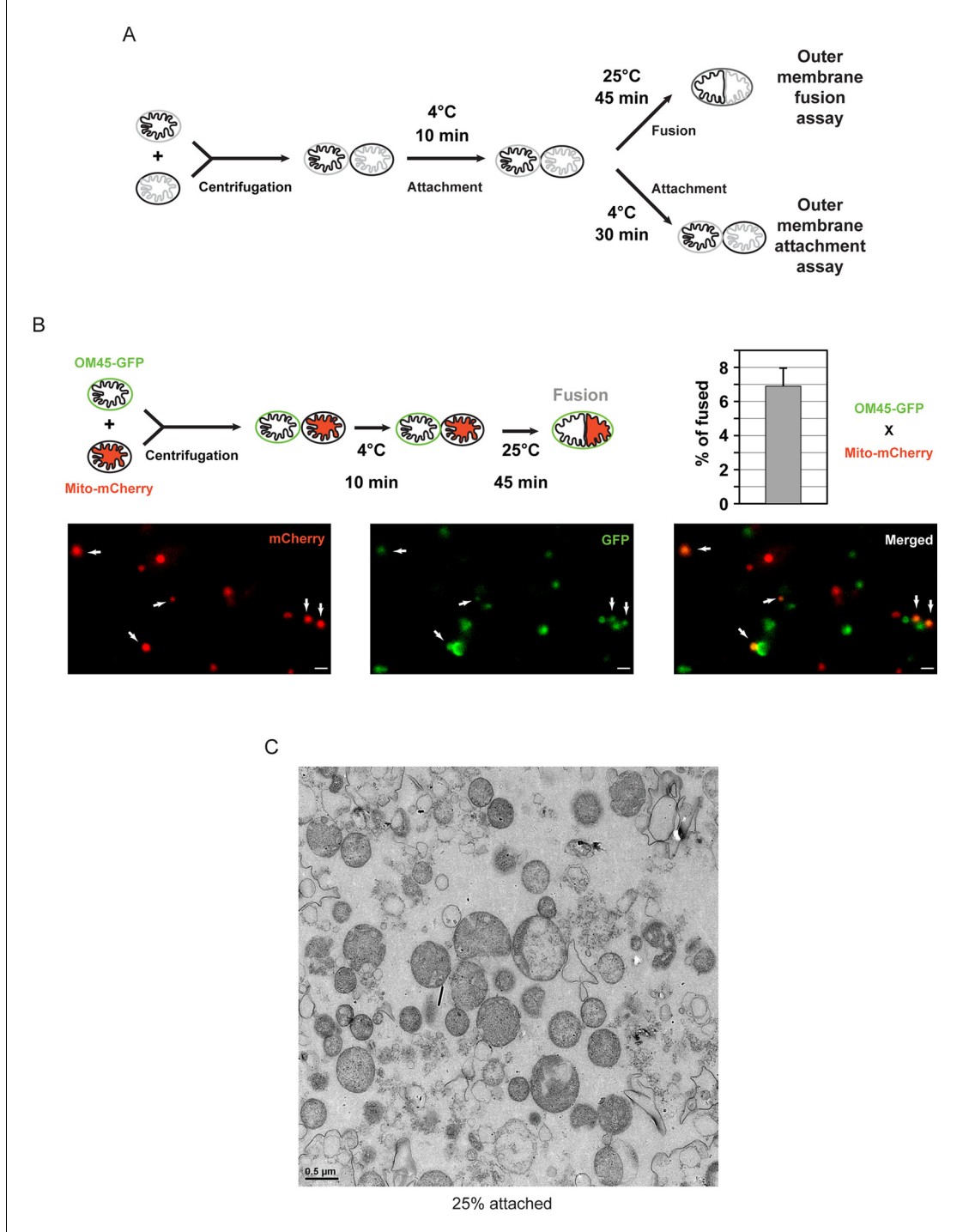

**Figure 1.** *In vitro* outer membrane fusion and attachment assays. (**A**) Purified mitochondria are brought into contact by centrifugation. A 10 min incubation on ice promotes mitofusin-dependent attachment, which is essential for subsequent fusion of outer membranes at room temperature (top). Prolonged incubation on ice prevents fusion of outer membranes but stabilizes attached intermediates (bottom). Upon incubation at 25°C, fusion of inner membranes does not occur unless energy is regenerated. (**B**) Top: Fusion reactions were performed by mixing mitochondria isolated from cells expressing either the outer membrane protein OM45 tagged with GFP (OM45-GFP) or the mitochondrial matrix targeted mCherry (Mito-mCherry). Bottom: Fluorescence microscopy of a fusion reaction. Co-localization of GFP and mCherry indicate intermediates with fused outer membranes (white arrows), scale bars 1 μm. Top right: Fusion efficiency. Error bar represents the s.d. from three independent experiments. (**C**) Representative transmission electron micrograph of *in vitro* attachment reactions with mitochondria isolated from wild-type cells.

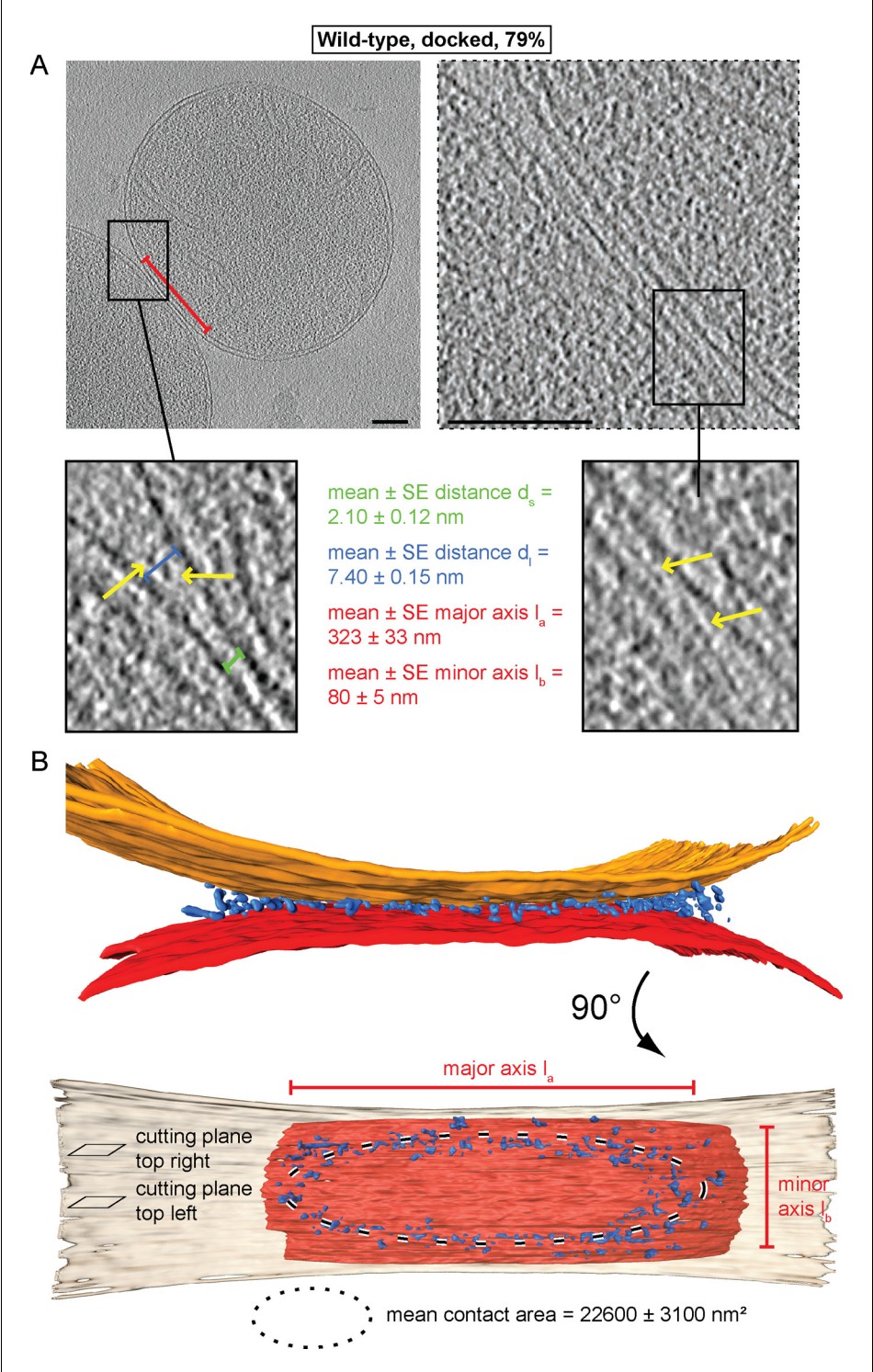

**Figure 2.** Cryo-ET of docked intermediates (79% of sampled wild-type mitochondria, see *Table 1*). (**A**) Slices and zooms through tomographic volumes at different z-heights (section planes indicated in B) through the center (left) or edge (right) of a contact area defined by its major axis, $l_a$ (red bracket), and the perpendicular minor axis $l_b$ (see B); scale bars 100 nm. Dense protein complexes (yellow arrows) are visible at a distance of 6–9 nm from the outer membrane ($d_l$, blue bracket) but not in the center of the contact area (1–3 nm, green bracket, $d_s$). Sections through the edge of a contact area (right) reveal interstitial densities between the outer membranes. (**B**) 3D rendering of outer membranes (red and orange) of two closely apposed mitochondria and protein densities around the contacts area (blue; the same color scheme is used in all figures).

*Figure 2 continued on next page*

*Figure 2 continued*

The following figure supplement is available for figure 2:

**Figure supplement 1.** Cryo-ET of major mitochondrial population of wild-type attachment intermediates (Docked, 79%, see *Table 1*).

be possible to visualize the contact sites and detect complexes that mediate mitochondrial attachment.

Cryo-ET revealed two main populations of attached mitochondria, characterized by morphologically distinct contacts between outer membranes. 79% of the contact areas were organized as regions where opposing outer membranes approached one another to a distance $d_s$ of 1–3 nm (*Figure 2A*, green bracket; *Figure 2—figure supplement 1A*), without observable density between the membranes. The contact areas displayed an average surface of 22,600 ± 3100 nm$^2$ with flattened outer membranes, prompting us to term this state 'docked' intermediates (*Figure 2A*). At the edges of the contact areas, where the distance $d_l$ between outer membranes reached 6–9 nm (*Figure 2A*, blue brackets; *Figure 2—figure supplement 1B*), we detected defined protein densities between the two outer membranes (*Figure 2A*; yellow arrows). Tomographic reconstruction (*Figure 2—figure supplement 1C*) and 3D rendered volumes (*Figure 2B*) of mitochondria revealed that contact areas are delimited by a dense ring-like structure, termed the docking ring (*Video 1*).

Contact areas of the remaining 21% of attached mitochondria were smaller (6900 ± 1700 nm$^2$) with an average distance of 6.7 ± 0.5 nm between outer membranes (*Figure 3*). In contrast to docked intermediates, there were clear densities between the apposed outer membranes over the complete contact area (*Figure 3A*; *Figure 3—figure supplement 1*). Frequently, this density appeared to consist of more or less regular repeats of globular protein units (*Figure 3B*; red arrows). These units, with an average spacing of 4.6 ± 0.7 nm (n = 13), occasionally extended two protrusions toward each membrane (*Figure 3B*; inserted zoom). We refer to these attached mitochondria as tethered intermediates, as they appeared to be tethered by proteins.

## GTP hydrolysis is required for formation of docking rings

The ring of densities (docked intermediates, *Figure 2B*) and the regular repeat of globular protein densities (tethered intermediates, *Figure 3*) indicate that protein complexes may be responsible for promoting mitochondrial attachment. As the homotypic attachment of outer membranes and their subsequent fusion depends on the GTPase activity of mitofusins (*Cohen et al., 2011*; *Koshiba et al., 2004*), we assessed the impact of GTP hydrolysis on the formation of the tethered and docked intermediates. Cryo-ET of *in vitro* attachment reactions treated with a non-hydrolysable GTP analog (GMP-PNP) revealed two distinct populations of attached mitochondria with different distances between their outer membranes (*Figure 4*).

In contrast to non-treated samples, only 14% of mitochondria formed contacts reminiscent of docked intermediates and with significantly smaller contact areas (*Table 1*). In the remaining 86% of attached mitochondria, the outer membranes were 6.3 ± 0.2 nm apart (*Figure 4A*, blue bracket). Densities were observed between apposed membranes with frequent repeats of 3–4 nm globular structures with, occasionally, two protrusions extending towards each membrane (*Figure 4A*, inserted zoom and *Figure 4B*, red arrows). These densities and their regular spacing (4.3 ± 0.6 nm; n = 24) were strikingly similar to those seen in wild-type tethered intermediates (*Figure 3B*). Moreover, among the attached intermediates we identified (*Table 1*), this subpopulation observed upon treatment with GMP-PNP displayed features that were essentially identical to the tethered intermediates seen in wild-type conditions both in terms of morphology (*Figure 4C*; *Figure 4—figure supplement*

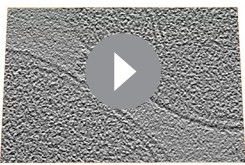

**Video 1.** 3D rendering of docked mitochondria as shown in *Figure 2*. Outer membranes in red and orange, distinct density in blue.

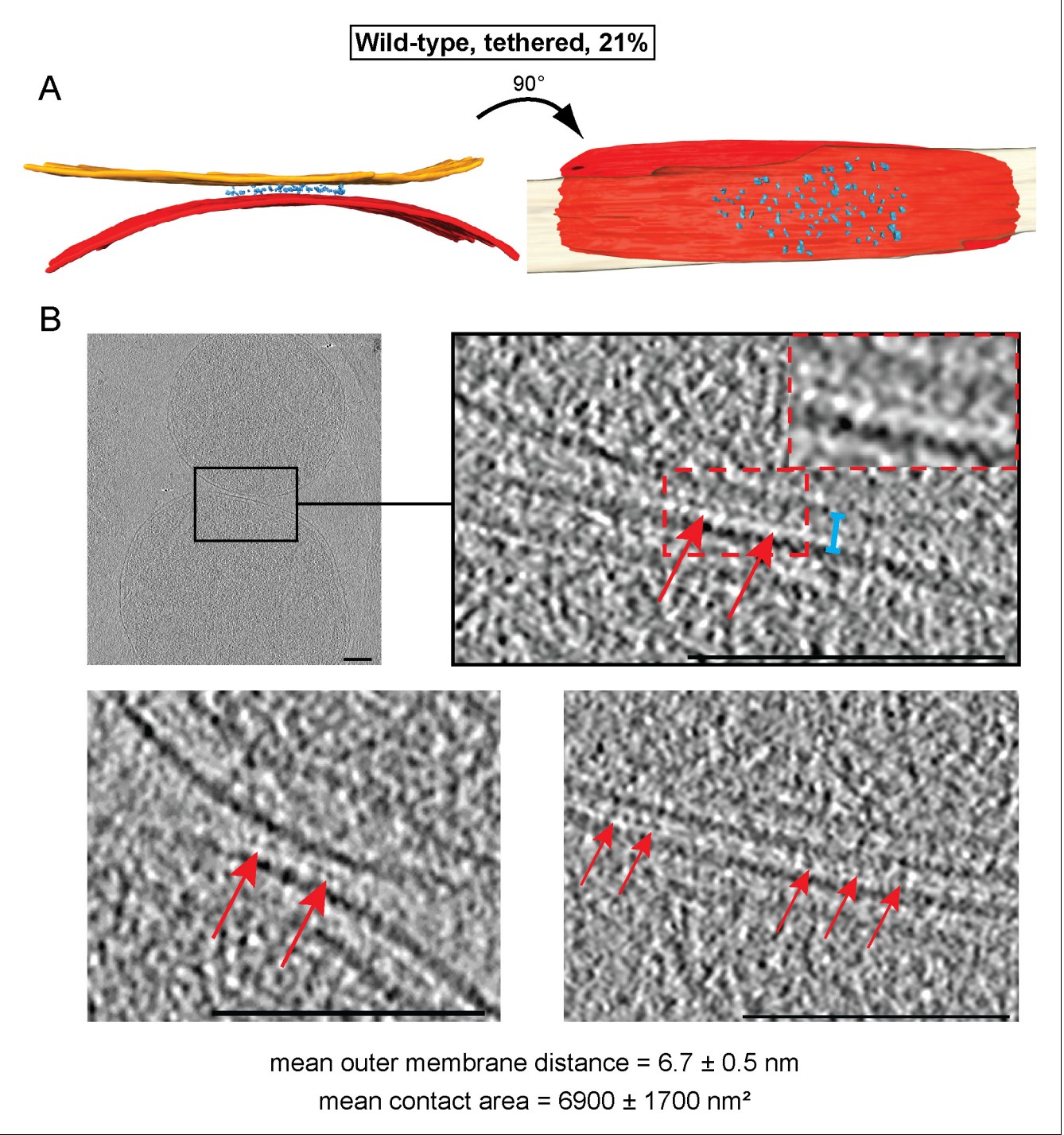

**Figure 3.** Cryo-ET analysis of tethered intermediates (21% of sampled wild-type mitochondria, see *Table 1*). (**A**) 3D rendering of two closely apposed mitochondria shown in B, top row and *Figure 3—figure supplement 1*. (**B**) Slices and zooms (indicated by black and red dashed boxes) through tomographic volumes; scale bars 100 nm. Blue bracket: outer membrane distance; red arrows, interstitial density.

The following figure supplement is available for figure 3:

**Figure supplement 1.** Cryo-ET analysis of the minor mitochondrial population of wild-type attachment intermediates (Tethered, 21%, see *Table 1*).

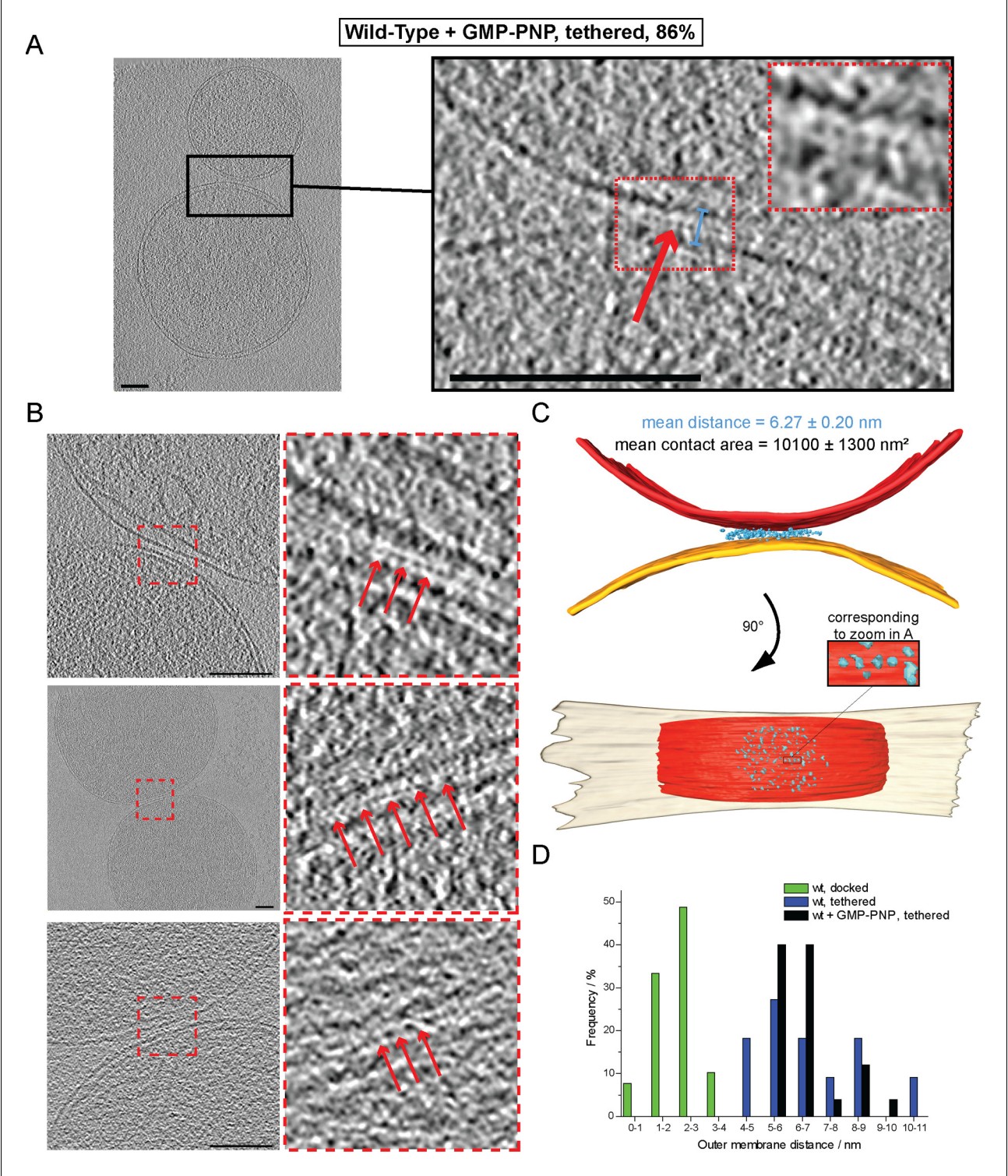

**Figure 4.** Cryo-ET of mitochondrial attachment intermediates upon GMP-PNP treatment (Tethered, 86%, see *Table 1*). (A-B) Example slices and zooms (black and red dashed boxes) through tomographic volumes; scale bars 100 nm; blue bracket: distance between outer membranes; red arrows: regularly spaced interstitial protein densities. Note the repeat distance of 4.3 nm. (C) 3D rendering of two closely apposed mitochondria shown in **A**. The zoomed densities correspond to the regularly spaced interstitial protein densities zoomed in **A**. (D) Histogram of distances between outer membranes for all major populations of attached intermediates.

The following figure supplement is available for figure 4:

*Figure 4 continued on next page*

*Figure 4 continued*

**Figure supplement 1.** Cryo-ET of a tethered mitochondrial intermediate upon GMP-PNP treatment (Tethered, 86%, see *Table 1*).

*1A*; *Video 2*) and statistics (*Figure 4D*; *Figure 4—figure supplement 1B–D*).

These observations suggest that inhibition of GTP hydrolysis induces the accumulation of tethered mitochondria at the expense of docked intermediates (21% tethered intermediates in wild-type conditions against 86% upon GMP-PNP treatment).

## GTP hydrolysis mediates the transition from globular protein repeats to docking rings

To find out if inhibition of GTP hydrolysis blocks formation of the docking ring at an early stage of the mitochondrial attachment process, we introduced GMP-PNP at distinct time points of the *in vitro* fusion reaction and analyzed the ratios of tethered and docked intermediates by cryo-ET (*Figure 5A*). Docked intermediates were predominantly found when GMP-PNP was added 10, 20 or 40 min after centrifugation (*Figure 5B*, blue squares), indicating that the ring had been already formed and that it was not dissolved by GMP-PNP. In contrast, addition of GMP-PNP prior to or immediately after centrifugation resulted in strong accumulation of tethered intermediates (*Figure 5B*, red circles), while docked intermediates were suppressed. Strikingly, addition of GMP-PNP two minutes after centrifugation resulted in equal proportions of tethered and docked intermediates (*Figure 5B*).

These observations demonstrate not only that tethering precedes docking, but also that transition from one state to the other is controlled by GTP hydrolysis. Consequently, the tethered intermediate with its repeating globular protein densities represents an early stage of the mitochondrial outer membrane fusion process. GTP hydrolysis then allows this stage to evolve towards the docked state with the docking ring.

Consistent with this, a discrete category of intermediates was detected among the mixed population of tethered and docked intermediates when GMP-PNP was added two minutes after centrifugation. This category was characterized by apposed membrane regions sandwiching protein densities identical to those in tethered intermediates and regions reminiscent of docked contact areas where outer membranes approached to within less than 3 nm (*Figure 5—figure supplement 1* and *Video 3*). This hybrid category of attached mitochondria probably represents a transition state from mitochondrial tethering towards mitochondrial docking.

## A final GTP hydrolysis step triggers the transition from docking to fusion

Of all attached mitochondria, docked intermediates would be the most suitable for outer membrane fusion. The large contact surface and close approach of outer membranes would poise them for fusion. The proportion of docked intermediates increases when outer membrane fusion can proceed but decreases when fusion is inhibited, as we showed by inhibiting GTP hydrolysis.

**Table 1.** Characteristics of mitochondrial attachment intermediates identified in this study. Abbreviations: o.e. overexpression, *i.a. inter alia*.

| Condition (n) | Contact type (n) | % | Densities organization | Distance ± SE / nm | Contact area ± SE / nm² |
|---|---|---|---|---|---|
| Wild-Type (52) | Docked (41) | 79 | Docking ring | 2.1 ± 0.1 | 22600 ± 3100 |
| | Tethered (11) | 21 | Interstitial/Few | 6.7 ± 0.5 | 6900 ± 1700 |
| Wild-Type + GMP-PNP (22) | Tethered (19) | 86 | Interstitial/Few | 6.3 ± 0.2 | 10100 ± 1300 |
| | Other (3) | 14 | *i.a.* Docking ring | 2.6 ± 0.3 | 11600 ± 3900 |
| Fzo1 O.E. (19) | Abortive (14) | 74 | Interstitial/ Numerous | 8.8 ± 0.4 | 5000 ± 900 |
| | Other (5) | 26 | Docking ring | 2.5 ± 0.4 | 4200 ± 1200 |

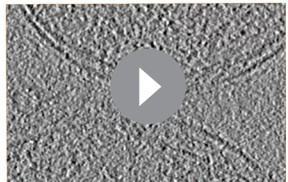

**Video 2.** 3D rendering of tethered mitochondria upon addition of GMP-PNP as shown in *Figure 4*. Outer membranes in red and orange, distinct density in blue. DOI: 10.7554/eLife.14618.012

The question of how docking promotes the formation of intermediates with fused outer membranes (*Figure 6—figure supplement 1*) was nonetheless puzzling. Fusion may occur anywhere within the contact area devoid of visible densities because this area corresponds to regions where membrane contact is closest. Alternatively, the docking ring might trigger fusion over the whole rim of the contact area, similar to the 'vertex ring' that assembles at the edge of docked vacuoles (*Wang et al., 2002*; *Wickner, 2010*). This ring, composed of SNAREs, small GTPases, tethering factors and lipid microdomains, promotes membrane fusion at the periphery of contact regions between vacuoles, generating a lumenal vesicle that degrades in the fused organelle. In the context of mitochondrial fusion, such an intralumenal outer membrane vesicle would block the subsequent attachment of inner membranes. Indeed, we did not find fused outer membrane intermediates of this kind.

However, close inspection of large populations of attached intermediates (from 87 high magnification micrographs) by Transmission Electron Microscopy (TEM) of stained plastic sections revealed a sub-category of mitochondria in close contact, presumably in the docked configuration, in which the outer membranes were fused near the rim of the contact region (*Figure 6—figure supplement 2A and B*). While the switch from docking to fusion is bound to be a rapid, transient process and cryo-ET can only sample specimen volumes of the order of $10^{-7}$ nanoliters, we succeeded in capturing such a docked intermediate. The tomographic volume shows that the outer membranes of two mitochondria were locally fused to form a toroid, 40 nm pore on one side of the contact area (*Figure 6A*). The toroid pore formed in the path of the docking ring (*Figure 6B* and *Video 4*). Densities in the vicinity of the fusion pore were sparse, suggesting that in this region the docking ring structure was in the process of disassembly (*Figure 6B*). These observations provide a proof of principle that the docked intermediates are competent for effective outer membrane fusion, raising the question of which molecular events trigger the transition from docking to outer membrane fusion.

To this end, we compared the effect of GMP-PNP on *in vitro* fusion efficiency either before tethering (*i.e.* after centrifugation; see *Figure 5B*) or after docking (*i.e.* after 10 min incubation on ice; see *Figure 5B*) (*Figure 6C*; see *Figure 1B*). Strikingly, the extent of outer membrane fusion impairment was similar (40 to 50%), irrespective of whether GMP-PNP was added at the beginning or at the end of the 10 min incubation period (*Figure 6D*). This indicates that it is not important whether the mitochondria were tethered or docked. Hence, GTP hydrolysis is not only required for the transition from tethering to docking but also for the transition from docking to fusion.

## Fzo1 overexpression inhibits mitochondrial docking

To further evaluate the functional relationship between mitofusins, mitochondrial docking and mitochondrial fusion, we took advantage of the fact that both the absence or the accumulation of mitofusins inhibits mitochondrial fusion *in vivo* (*Cohen et al., 2011*; *Escobar-Henriques et al., 2006*; *Koshiba et al., 2004*). Consistent with this, absence or 50-fold overexpression of Fzo1 (*Figure 7A*; *fzo1△* and *FZO1 o.e.*) totally abolished respiratory growth at 30℃ (*Figure 7B*).

In the absence of Fzo1, cryo-ET analysis revealed small mitochondria well separated from each other but mitochondrial attachment was not detected (*Figure 7—figure supplement 1A*), which is consistent with the essential function of mitofusins in mitochondrial anchoring (*Cohen et al., 2011*; *Koshiba et al., 2004*). On the other hand, the fusion defect caused by the overexpression of Fzo1 may result from the accumulation of Fzo1 molecules on outer membranes and an imbalance with other proteins implicated in mitochondrial fusion, such as Ugo1 and Mgm1 (*Figure 7—figure supplement 1B*). Notably, Fzo1 overexpression also induces a specific phenotype of mitochondrial aggregation (*Figure 7C*) in which mitochondrial puncta aggregate in one region of the cell cortex (*Figure 7—figure supplement 1C*). This phenotype suggests that, as well as inhibiting fusion, overexpression of Fzo1 may also promote the attachment of mitochondria to each other.

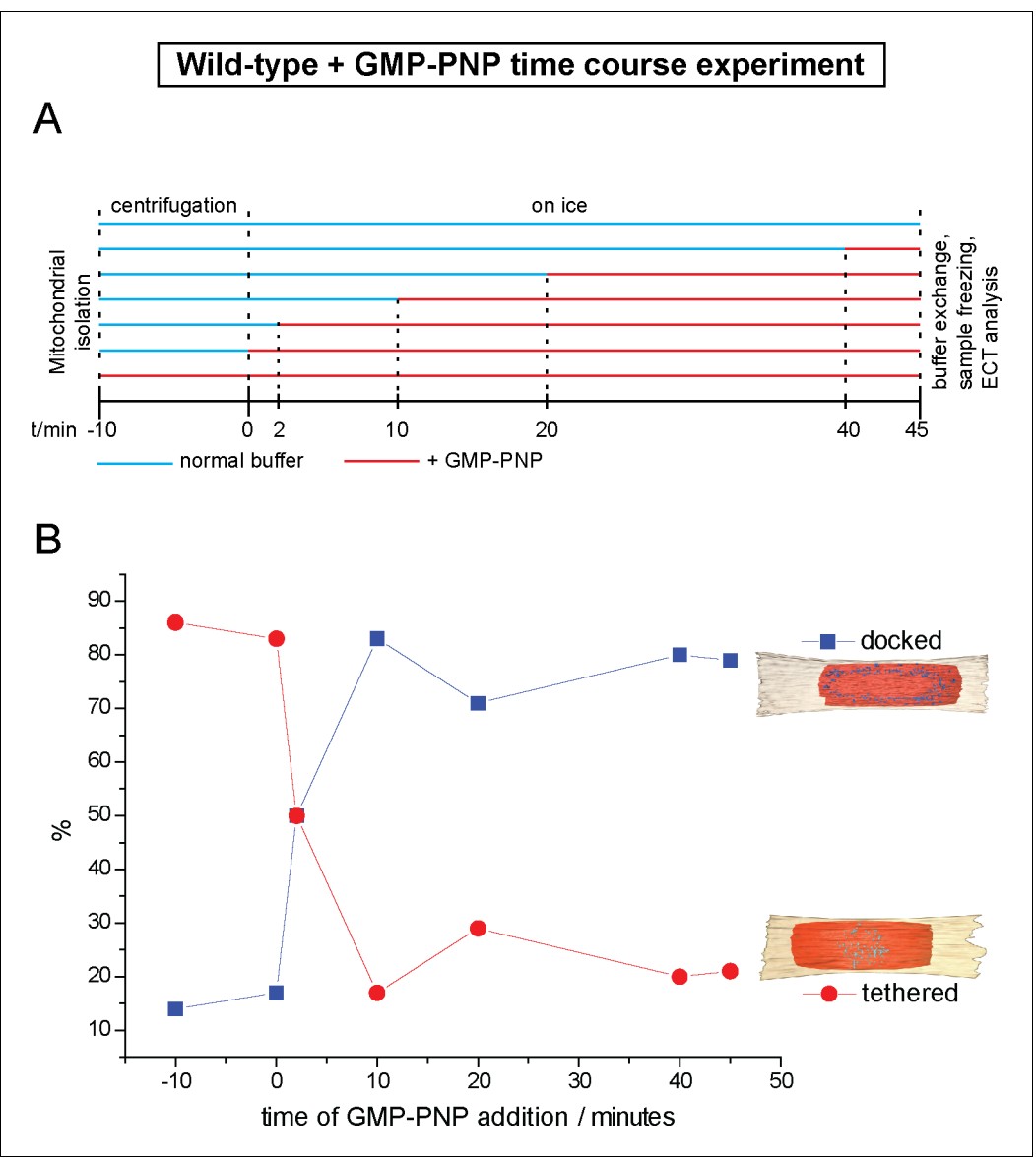

**Figure 5.** Cryo-ET time course experiment of attached mitochondria treated with GMP-PNP. (**A**) GMP-PNP was added at the time points indicated (dashed black lines). (**B**) Proportion of docked and tethered intermediates observed before and after GMP-PNP addition. Time t = 0 indicates the start of the incubation period on ice.

The following figure supplement is available for figure 5:

**Figure supplement 1.** Hybrid intermediate captured upon GMP-PNP addition two minutes after centrifugation.

To verify this, *in vitro* fusion assays were performed using mitochondria isolated from cells overexpressing Fzo1. As expected, *in vitro* outer membrane fusion was strongly inhibited but mitochondrial attachment was significantly increased, with 50% of mitochondria attached to others (*Figure 7D* and *Figure 7—figure supplement 1D*). Consistent with their compromised fusing ability, mitochondria from Fzo1-overexpressing cells were significantly smaller than controls. Further analysis indicated that Fzo1 overexpression induced a two-fold increase in mitochondrial attachment both before and after centrifugation (*Figure 7—figure supplement 1E–F*). Thus, mitochondria purified from Fzo1-overexpressing cells retained an increased attachment capacity, indicating that the *in vivo* aggregation phenotype (*Figure 7C*) is caused, at least in part, by perturbations that are intrinsic to outer membranes.

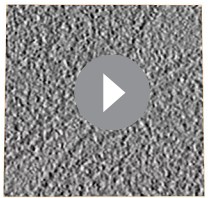

**Video 3.** 3D Rendering of tethered intermediate upon addition of GMP-PNP after two minutes, as shown in *Figure 5—figure supplement 1*. Outer membranes in red and orange, distinct density in blue.

To better characterize these perturbations, attached mitochondria were analyzed by cryo-ET. While intermediates with a docking ring were formed (*Figure 7—figure supplement 2A–C*), they were observed less frequently (26%) and the average contact surface was significantly smaller compared to docking intermediates observed under wild-type conditions (*Table 1*). In the remaining attached intermediates (74%), the surface of apposition was also reduced (4970 ± 930 nm$^2$), with outer membranes on average 8.8 ± 0.4 nm apart (*Figure 7E*; blue bracket; *Figure 7—figure supplement 2D–E*). Importantly, numerous densities accumulated in regions of closest contact between mitochondria (*Figure 7F*), but the densities were disorganized and did not form globular protein repeats (*Figure 7E–F*). These attached mitochondria, which were clearly distinct from tethered intermediates observed under wild-type conditions, likely correspond to artefactual and abortive fusion intermediates.

These results indicate that while normal levels of Fzo1 are required for the formation of *bona fide* docking rings, overexpression of the mitofusin induces the formation of protein aggregates that perturb the regulated sequence of events required to reach productive mitochondrial docking.

## Fzo1 enrichment at mitochondrial contact sites

Densities detected at the junctions of tethered and docked mitochondria correspond to protein complexes responsible for promoting productive attachment of outer membranes prior to their fusion. Several protein factors either within or extrinsic to the mitochondrial outer membrane may assemble into such complexes (*Coonrod et al., 2007*; *Hoppins et al., 2009*; *Sesaki and Jensen, 2001*; *2004*). Notably, the units with a defined central density and protrusions extending towards each lipid bilayer bridged the apposed outer membranes (*Figure 8A*). These units were stabilized in the presence of GMP-PNP (*Figure 4*) and their arrangement may reflect the self-association in *trans* of a factor extruding from outer membranes (*Figure 8B*). These features thus point to a transmembrane GTPase specialized in connecting outer membranes, raising the possibility that the observed units are indeed *trans*-oligomers of Fzo1 molecules.

The presence and active involvement of mitofusins in mitochondrial tethering and docking is not only consistent with the established function of these DRPs in attachment and fusion of outer membranes (*Cohen et al., 2011*; *Hermann et al., 1998*; *Ishihara et al., 2004*; *Koshiba et al., 2004*; *Legros et al., 2002*; *Shutt et al., 2012*) but also with their documented accumulation at mitochondrial junctions (*Hoppins et al., 2009*). To validate Fzo1 as a potential component of the densities found at mitochondrial contact sites, we thus devised an *in situ* protein labeling strategy for cryo-ET. Recent experiments to label mitochondria for cryo-ET with a purified, biotin-labelled protein import substrate conjugated with streptavidin-coated Quantum Dots have been successful (*Gold et al., 2014*). To biotinylate mitofusin molecules *in situ*, Fzo1 was C-terminally fused to the Avi tag, a 15 amino-acid peptide that is recognized and can undergo specific biotinylation by the *E. coli* biotin ligase BirA (*Beckett et al., 1999*; *van Werven and Timmers, 2006*). Mitochondria isolated from wild-type (*FZO1*) or *FZO1-Avi* cells were biotinylated *in vitro* with recombinant BirA before processing for outer membrane attachment assays, followed by incubation with streptavidin-coupled Q-Dots to label the Fzo1-Avi construct (*Figure 8—figure supplement 1*). While Q-Dots were rarely found on the surface of tagged or untagged mitochondria (*Figure 8C and D*, right graph), they were enriched at the junction of attached *FZO1-Avi* mitochondria as compared to attached *FZO1* mitochondria (*Figure 8D*, left graph). Moreover, Q-Dots, and therefore the *FZO1-Avi* constructs, were located either at the periphery of docked contact sites or between tethered contact sites (*Figure 8E*). These data confirm the accumulation of mitofusins at mitochondrial contact sites and corroborate the likely contribution of Fzo1 to densities found at junctions of tethered and docked intermediates.

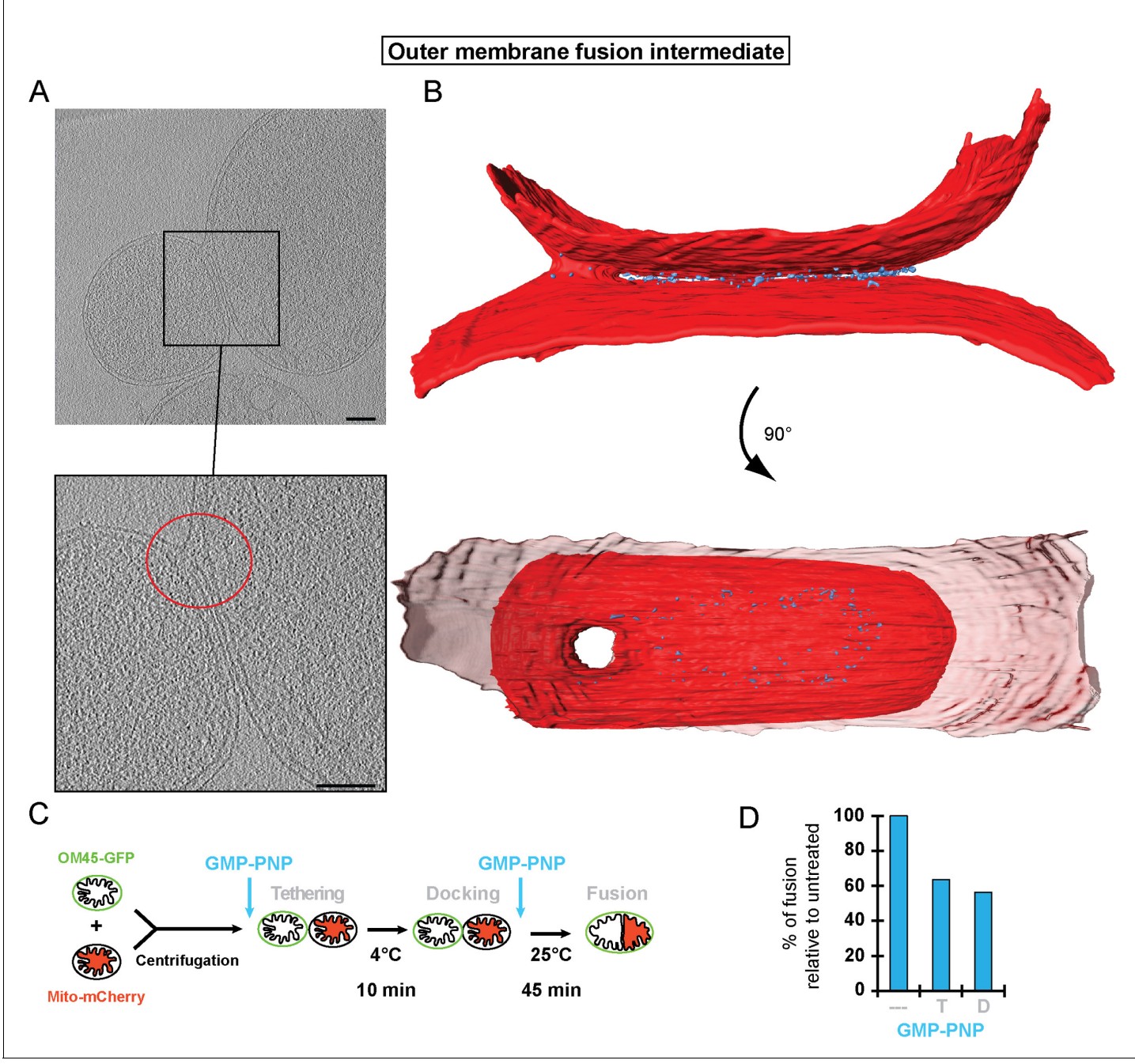

**Figure 6.** Stages of outer membrane fusion. (**A** and **B**) Cryo-ET of docked mitochondria with partially fused outer membrane. (**A**) Slices through tomographic volume; scale bars 100 nm. The red circle highlights the region of outer membrane fusion. (**B**) 3D rendering. Two mitochondria are joined by one continuous outer membrane (red). The inter-membrane spaces are connected by a toroid pore of 40 nm diameter. (**C** and **D**) Fluorescence microscopy of *in vitro* outer membrane fusion. GMP-PNP was added at the beginning (tethering, T) or at the end (docking, D) of the 10 min incubation period.

The following figure supplements are available for figure 6:

**Figure supplement 1.** Intermediate with fully fused outer membrane.

**Figure supplement 2.** Intermediates with partially fused outer membranes.

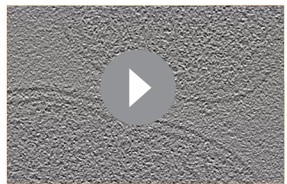

**Video 4.** 3D rendering of partially fused mitochondria as shown in *Figure 6*. Outer membranes in red, distinct density in blue.

## Discussion

Prior to this study, the initial steps of mitochondrial outer membrane fusion were known to involve mitofusins and to depend on GTP-binding and hydrolysis (*Cohen et al., 2011*; *Escobar-Henriques and Anton, 2013*; *Ishihara et al., 2004*; *Koshiba et al., 2004*; *Shutt et al., 2012*). Combining an *in vitro* mitochondrial attachment assay with cryo-ET, we were able to dissect this process into several distinct steps. Our experiments on the inhibition of GTP hydrolysis allow us to temporally position each intermediate observed and thus provide the basis for a refined and comprehensive model of mitochondrial outer membrane fusion (*Figure 9*).

## Mitochondrial outer membrane attachment and fusion

Initially outer membranes of two attached mitochondria are tethered by globular protein repeats and membranes approach to within 6 nm (*Figure 9*, step 1). These tethered intermediates are seen in wild-type conditions (*Figure 3*) and accumulate upon addition of GMP-PNP (*Figure 4*). Subsequently, GTP hydrolysis allows the fusion process to evolve progressively towards mitochondrial docking (*Figure 5*), as evidenced by the hybrid intermediates observed upon addition of GMP-PNP two minutes after initiation of tethering (*Figure 5—figure supplement 1*). Consistent with this, GTP hydrolysis by *trans*-complexes of atlastins precedes vesicle fusion and is essential to promote tethering of proteoliposomes (*Liu et al., 2015*; *Saini et al., 2014*).

The docked intermediates are characterized by a docking ring of protein density that surrounds extended areas with outer membranes separated by less than 3 nm and devoid of visible densities (*Figure 9*, step 2). The capture of docked intermediates with partly fused outer membranes, the location of the fusion pore in the path of the docking ring undergoing disassembly and the inhibition of fusion upon treatment of docked mitochondria with GMP-PNP, demonstrates that docking is the stage that precedes merging of outer membranes (*Figure 6*). We do not exclude that small protein complexes, which are not visible by cryo-ET, might reside between outer membranes and participate in the fusion process. However, our observations do suggest that the docking ring of protein densities is the driving force for subsequent bilayer merging. We thus propose that the fusion of bilayers is initiated by further GTP hydrolysis in the path of the docking ring where the outer membrane curvature is most pronounced (*Figure 9*, step 3). This step may trigger the disassembly of the docking ring. Tethering (*Figure 9*, step 4) and fusion of the inner membrane (*Figure 9*, step 5) mediated by OPA1/Mgm1 then completes the mitochondrial fusion process.

## Mitofusins are involved in tethering, docking and fusion of outer membranes

Taking into account the essential role of mitofusins in mitochondrial attachment and fusion (*Hermann et al., 1998*; *Koshiba et al., 2004*), we obtained several lines of evidence that Fzo1 contributes to the formation of the macromolecular assemblies we discovered at mitochondrial junctions.

Overexpression of Fzo1 inhibited mitochondrial docking and fusion but stimulated the formation of artefactual tethering intermediates that are characterized by an accumulation of protein aggregates at mitochondrial junctions (*Figure 7*). This result demonstrates once more that absence of docking correlates with deficient mitochondrial fusion and implies that normal levels of mitofusins are required for the formation of the docking rings. It is also important to realize that increased levels of mitofusins correlate with the accumulation of protein densities at mitochondrial junctions. However, whether these aggregates correspond to abortive mitofusin oligomers remains speculative.

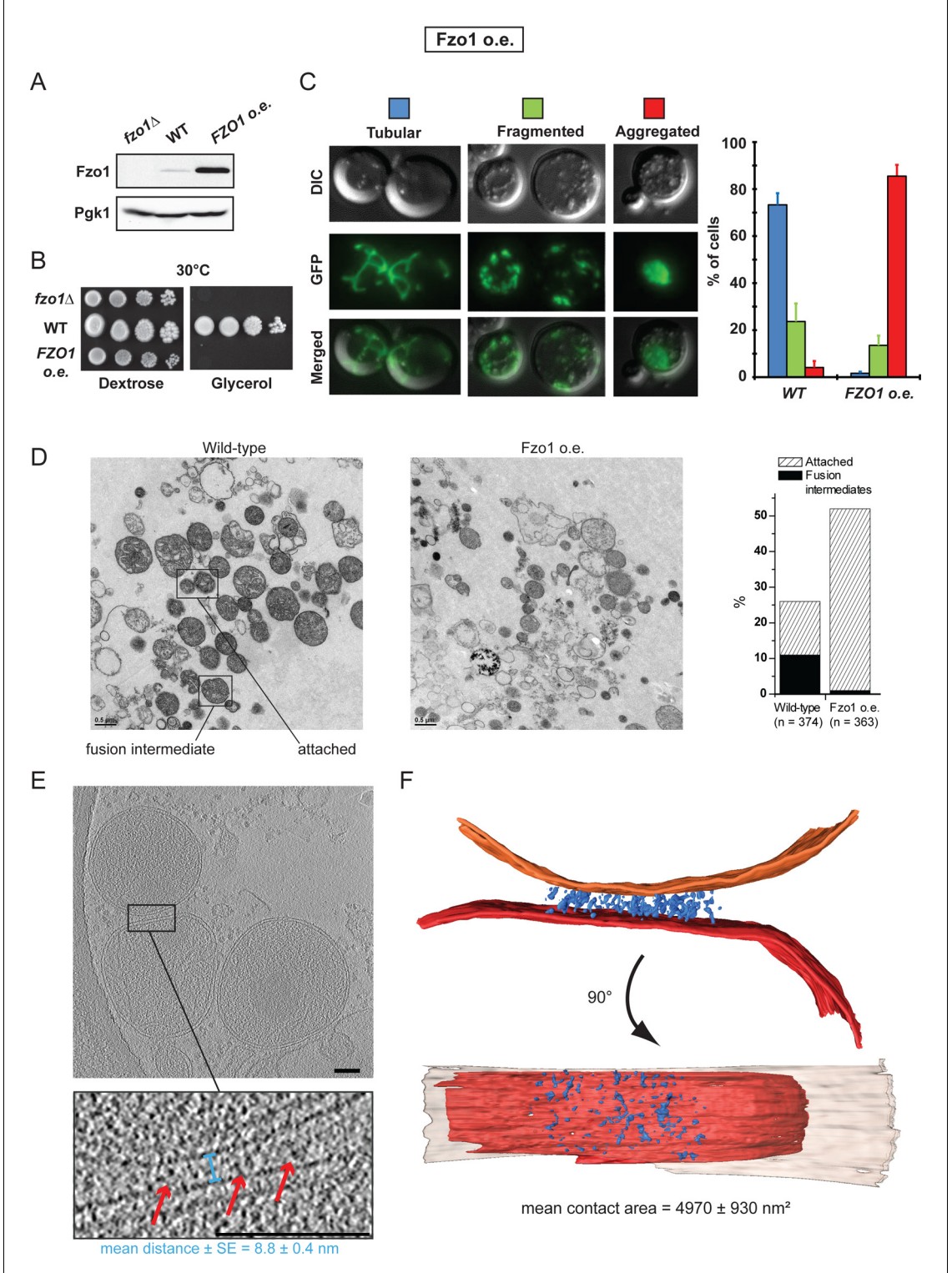

**Figure 7.** Overexpression of Fzo1. (**A**) Total protein extracts of *fzo1Δ* cells transformed with an empty vector (*fzo1Δ*), pRS314-FZO1 (*WT*) or pRS414-TEF-FZO1 (*FZO1 o.e.*) were analyzed by anti-Fzo1 and anti-Pgk1 immunoblotting. Fzo1 is overexpressed about 50 fold in *FZO1 o.e.* as compared to *WT* conditions. (**B**) Serial dilutions of cells from **A** grown in the presence of glucose or glycerol as the sole carbon source at 30°C. Lack or overexpression of Fzo1 both abolishes respiration and, therefore, growth on glycerol, consistent with inhibition of mitochondrial fusion. (**C**) Mitochondrial morphology in *WT* and *FZO1 o.e.* cells. Left: Representative morphologies. Right: Percentage of *WT* and *FZO1 o.e.* cells with indicated mitochondrial morphologies. Error bars represent the s.d. from three independent experiments. (**D**) Left: TEM analysis of *in vitro* outer membrane fusion reactions performed with

*Figure 7 continued on next page*

*Figure 7 continued*
mitochondria isolated from wild-type cells or cells overexpressing Fzo1. Note that mitochondria from Fzo1 o.e. cells are smaller than from wild-type cells. Right: Effect of Fzo1 overexpression on outer membrane fusion and attachment *in vitro*. (E) Slices through tomographic volume of mitochondrial attached intermediates upon Fzo1 overexpression (abortive, 74%, see *Table 1*); outer membrane distance (blue bracket) and densities between outer membranes (red arrows) are indicated. (F) 3D rendering of two closely apposed mitochondria shown in E.
The following figure supplements are available for figure 7:

**Figure supplement 1.** Absence or accumulation of Fzo1.
**Figure supplement 2.** Cryo-ET of mitochondria from Fzo1-overexpressing cells.

GMP-PNP that may bind to mitofusins and inhibit their GTPase activity (*Amiott et al., 2009*; *Ishihara et al., 2004*), induced the accumulation of tethered intermediates and prevented progression towards the docked stage (*Figure 4*). Moreover, Fzo1 enrichment at mitochondrial contact sites was confirmed by Q-dot labeling (*Figure 8*). To our knowledge, no other GTPase has been shown to be involved in outer membrane fusion. In this context, our observations converge to propose that Fzo1 is at least a component of globular protein repeats and docking rings, possibly together with other, as yet unknown factors.

In fact, the units composed of a central density with protrusions extending toward each outer membrane suggest that they may be Fzo1 *trans*-oligomers (*Figure 8A*). So far, the only structural insight into mitofusins derives from the HR2 domain of Mfn1 that has been proposed to tether outer membranes at a distance of 16 nm (*Koshiba et al., 2004*). This would exceed the 6 nm membrane spacing we measured in tethered intermediates. The overall morphology of the HR2 dimer can hardly account for the distinctive shape of the protein units we observed. However, the structure of the bacterial Fzo1 homolog BDLP in the open conformation (*Low and Löwe, 2006*; *Low et al., 2009*) and the established *trans*-interaction of atlastins through their GTPase domain (*Klemm et al., 2011*; *Byrnes et al., 2013*; *Liu et al., 2015*; *Saini et al., 2014*) allow us to propose an alternative model. Similar to BDLP, mitofusins bound to GMP-PNP or GTP may adopt an open conformation that, in analogy to atlastins, would promote *trans*-oligomerization of Fzo1 molecules through their GTPase domain. Conformational changes of the mitofusin oligomers upon GTP hydrolysis would pull the outer membranes closer together. This model is not only consistent with the shape of the protein units we visualized but also with the essential requirement of GTP hydrolysis for transition from mitochondrial tethering to mitochondrial docking we unraveled in this study. Hence, the structure of mitofusins with or without bound GTP will be instrumental to evaluate this model.

### *In vivo* vs *in vitro* mitochondrial outer membrane fusion

An absolute prerequisite for mitochondrial fusion *in vivo* is that the tips of mitochondrial tubules come close enough to promote attachment between outer membranes. In yeast cells, this requires the participation of actin filaments (*Simon et al., 1995*; *Smith et al., 1995*). In the *in vitro* fusion system, the essential role of the cytoskeleton is replaced by centrifugation to bring mitochondria into close enough contact for fusion to proceed (*Meeusen et al., 2004*). However, centrifugation is not sufficient, and an incubation of at least 10 min on ice, previously suggested to promote Fzo1 association in *trans*, has been shown to be essential for *in vitro* fusion (*Meeusen et al., 2004*).

Our results reveal that incubation on ice promotes the transition from tethering to docking, which is an active process that requires several rounds of GTP hydrolysis to progressively bring opposing outer membranes closer over an extended surface area but is not sufficient for their effective fusion. The observation that fusion starts at the edge of the docking ring and also depends on GTP hydrolysis suggests that the transition from tethering to docking brings membranes closer over an extended area. This would induce a locally increased curvature of the lipid bilayer, which may be critical. An ultimate cycle of GTP hydrolysis in this region of local membrane curvature would therefore lead to fusion instead of bringing membranes closer together. Notably, micrographs taken from gastric mucosa of a mole, featured in Chapter 7 of Don W. Fawcett's 'The Cell', present a series of three pairs of mitochondria proposed to represent successive stages of mitochondrial fission (*Fawcett, 1981*). The middle stage intermediate from this series and the fusion events identified in our

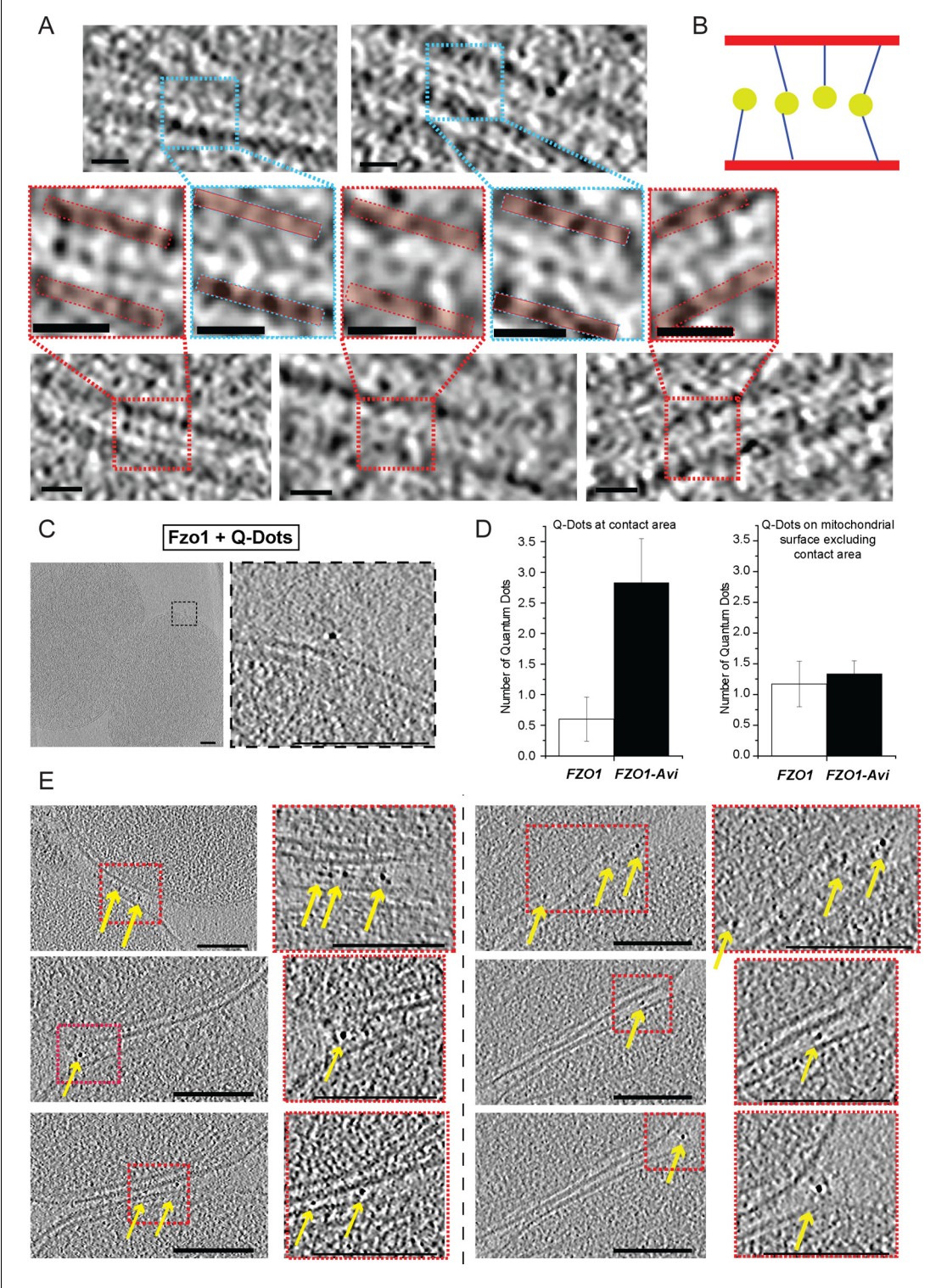

**Figure 8.** Fzo1 enrichment at mitochondrial junctions. (A) Cryo-ET of tethered mitochondria. Slices through tomographic volumes and zooms on contact regions (red and blue boxes); Outer membranes on zooms are delimited by red bars; scale bars 10 nm. (B) Scheme representing the central density (yellow circle) and extensions (blue lines) to outer membranes (red bars) for densities detected at the junction of tethered intermediates. (C–E) Q-Dot labelling of *FZO1* and *FZO1-AVI* mitochondria. (C) Control with non-labelled Fzo1. Tomographic slice and zoom with Q-Dots on the outer membrane. (D) Q-Dots per contact area (left) or on mitochondrial surface excluding the contact area (right) for *FZO1* (white) and *FZO1-Avi* (black). (E)

*Figure 8 continued*

Slices through tomographic volumes of Fzo1-Avi mitochondria labelled with Q-Dots (yellow arrows). Zooms are indicated by dashed red boxes. Scale bars 100 nm.

The following figure supplement is available for figure 8:

**Figure supplement 1.** Biotinylation and Q-Dot labelling of Fzo1-Avi.

study actually appear strikingly similar. This not only suggests that the sequence of events shown in this chapter might in fact correspond to successive stages of fusion but also that our model of outer membrane fusion may apply *in vivo* with mitochondria from mammalian cells.

Whereas *in vitro* local membrane deformation would be promoted exclusively by successive cycles of GTP hydrolysis after mitochondria become tethered through centrifugation, *in vivo* the transition from tethering to docking would also involve the action of the cytoskeleton. In the cell, the process of mitochondrial fusion as indicated by our *in vitro* studies would thus be regulated by cytoskeletal factors.

Regulation of mitochondrial fusion *in vivo* also involves post-translational modification of mitofusins (*Anton et al., 2011*; *Cohen et al., 2008*; *Shutt et al., 2012*). In yeast, the efficient fusion of outer membranes requires Fzo1 ubiquitylation by the Mdm30 ubiquitin ligase and subsequent degradation by the proteasome (*Cohen et al., 2008*). While its precise function is yet to be fully characterized, this regulation was previously shown to take place at the stage of mitochondrial attachment (*Cohen et al., 2011*). It is therefore conceivable that the UPS-dependent regulation of Fzo1 participates in regulating proper assembly of docking rings, which is consistent with the observation that high levels of Fzo1 inhibit mitochondrial docking (*Figure 7*). At this stage, we cannot exclude that ubiquitylation and degradation of Fzo1 regulates the transition from docking to effective fusion of outer membranes. Further investigations will be required to answer the fascinating question of how mitochondrial fusion is regulated.

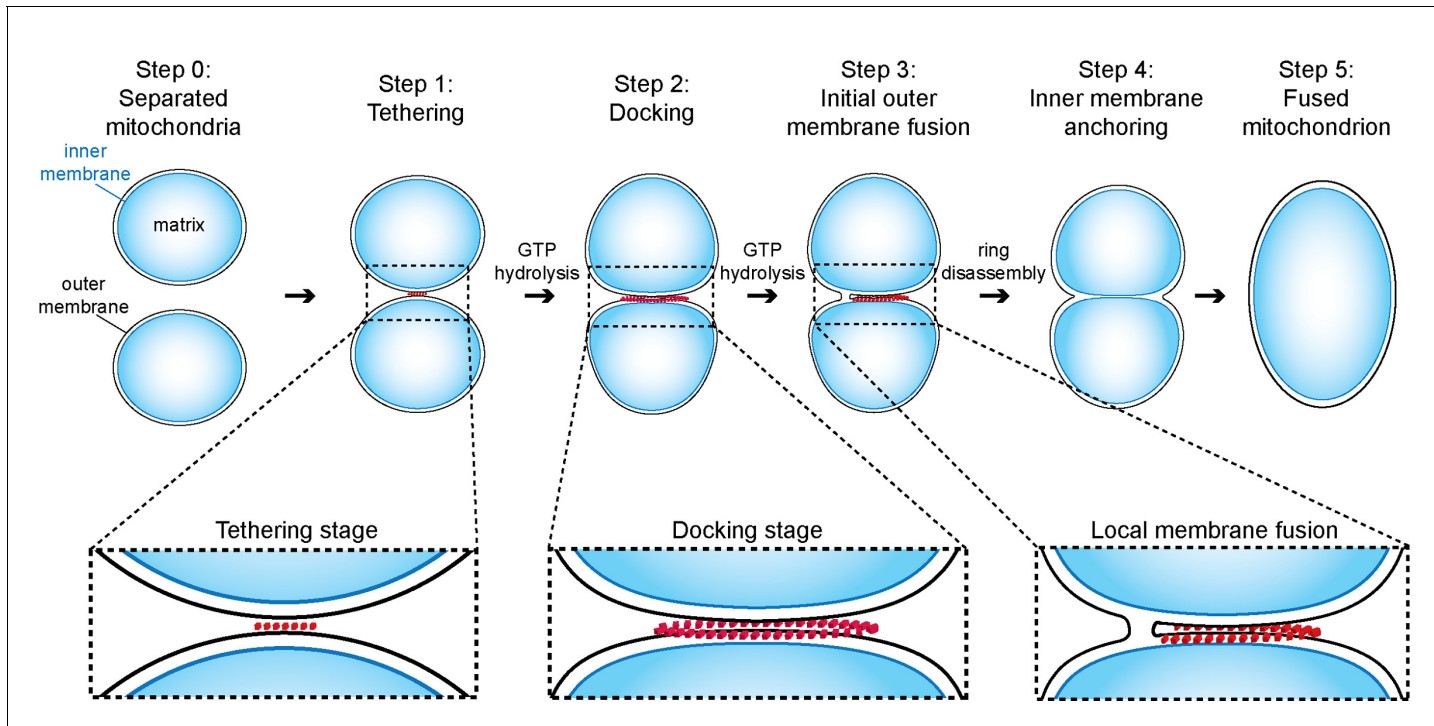

**Figure 9.** Model of outer membrane tethering, docking and fusion. Mitochondria (blue) are tethered or docked by protein complexes (red).

## Conclusion

Our work provides a detailed dissection of the outer mitochondrial membrane fusion process *in vitro* and highlights the crucial involvement of Fzo1 in this system. Similar mechanisms involving atlastins or OPA1/Mgm1 are likely to be active in the fusion of ER and mitochondrial inner membranes, respectively. Future challenges include deciphering the precise composition of the complexes mediating outer membrane fusion, and dissecting the processes regulating their formation and function.

# Materials and methods

### Yeast strains, plasmids and growth conditions

The *S. cerevisiae* strains and plasmids are listed in *Supplementary file 1*. Standard methods were used for growth, transformation and genetic manipulation of *S. cerevisiae*. Complete media and minimal synthetic media [Difco yeast nitrogen base (Voigt Global Distribution Inc; Lawrence, KS), and drop-out solution] supplemented with 2% dextrose (YPD and SD) or 2% glycerol (YPG and SG) were prepared as described (*Sherman et al., 1986*).

### Mitochondrial enriched preparations

Mitochondrial fractions for *in vitro* attachment and fusion were prepared as previously described (*Ingerman et al., 2007*). Cells were cultured to stationary phase in dextrose medium and then shifted to glycerol medium (or fresh dextrose medium for cells affected in respiration) to grow to a final OD 0.8–1.0. Cell walls were disrupted by incubation at 30°C with 100 mM Tris-HCl pH 9.4 and 50 mM β-mercaptoethanol for 20 min and subsequently with 1.2 M sorbitol plus zymolyase (Zymo Research; Irvine, CA) for 30 min. The resulting spheroplasts were lysed in cold NMIB (0.6 M sorbitol, 5 mM $MgCl_2$, 50 mM KCl, 100 mM KOAc, 20 mM Hepes pH 7.4) by douncing. After centrifugation at 4°C of the lysate at 3000 x g for 5 min, the supernatant was further centrifuged at 10170 x g, 4°C for 10 min to yield a pellet enriched in mitochondria. Protein concentration in mitochondria-enriched fractions was determined by Bradford assay (Bio-Rad Protein Assay; Bio-Rad Laboratories GmbH, Germany).

### *In vitro* mitochondrial attachment/fusion assays

Homotypic and heterotypic attachment/fusion reactions were respectively carried out with 0.5 mg of type 1 mitochondria or by mixing 0.25 mg of type 1 mitochondria with 0.25 mg of type 2 mitochondria. Mitochondria were then brought in vicinity by centrifugation at 10170 x g, 4°C for 10 min. Pellets were left on ice for 10 min before the supernatant was replaced by Stage 1 buffer (20 mM Pipes–KOH pH 6.8, 150 mM KOAc, 5 mM $Mg(OAc)_2$, 0.6 M sorbitol). Mitochondria were then left on ice for 30 min in attachment assays or incubated at 25°C for 45 min in outer membrane fusion assays. Resulting attachment and fusion reactions were subsequently processed for TEM, cryo-ET or fluorescence microscopy analysis.

### Miscellaneous variations during *in vitro* mitochondrial attachment/fusion reactions

Depending on experiments, discrete variations in attachment/fusion reactions described above were used. To assess the effect of GMP-PNP on attachment of wild-type mitochondria (*Figure 4*), GMP-PNP (1.5 mM, Sigma-Aldrich; St Louis, MO) was added during mixing of mitochondria (prior centrifugation) and was kept at constant concentration during the whole attachment reactions (including in Stage 1 buffer). In contrast, 6 mM GMP-PNP were added where indicated in time course experiments (*Figure 5*). Similarly, 12 mM GMP-PNP were added at tethering or docking stages in outer membrane fusion reactions shown in *Figure 6C*. To evaluate effects of Fzo1 overexpression on attachment and fusion, *in vitro* assays were performed with mitochondria-enriched fractions prepared from wild-type (MCY553) or Fzo1-overexpressing cells (MCY1222) grown in dextrose medium.

### Transmission electron microscopy

For analysis of attachment and outer membrane fusion intermediates by TEM, the fusion reactions were mixed with 20 volumes of fixative solution (3% paraformaldehyde, 1.5% glutaraldehyde, 2.5%

sucrose, 100 mM sodium cacodylate, pH 7.4) for 20 hr and subsequently washed twice with 100 mM sodium cacodylate, pH 7.4. To improve contrast, samples were post-fixed with 1% osmium tetroxide, 100 mM sodium cacodylate, pH 7.4, for 1 hr, after which samples were washed in distilled water. Then samples were embedded in 4% agar and washed with 50 mM acetate buffer pH 5.2 before they were stained with 1% uranyl acetate overnight (12 h) at 4°C. For plastic embedding, the samples were dehydrated in an ethanol gradient series (1 × 20 min 30%, 2 × 20 min 50%, 2×30 min 70%, 2 × 30 min 90%, 1 × 60 min 100% ethanol) followed by a switch to 1,2 propylenoxid (3 x 20 min 100%) and subsequently infiltrated using the Low Viscosity Premix Kit-Medium (Agar Scientific, England; 2 × 20 min 30%, 2 × 30 min 50%, 2 × 30 min 75%, overnight 100%, 2 × 2 h 100%). Polymerization was carried out at 65°C for 16 h. Thin sections (60–70 nm) were prepared with an Ultracut S microtome (Reichert, Germany), collected on 100 mesh copper grids coated with Pioloform FN 65 (Wacker Polymer Systems GmbH, Germany) and were double stained with 2% uranyl acetate for 2 min and lead citrate for 1 min. Sections were inspected with a transmission electron microscope (EM 208S; FEI, Germany) at 80 kV equipped with 2 x 2 k CCD camera (Gatan, Inc; Pleasanton, CA). Ratios of attached and fused mitochondria were obtained by dividing the number of mitochondria in physical contact or those with fused outer membranes over the total number of mitochondria (n > 300). The fused intermediates shown in *Figure 6—figure supplement 1* and *2* were obtained from the analysis of 87 high magnification micrographs containing one pair of attached or fused intermediates each.

## Electron cryo-tomography

For cryo-ET, mitochondria were washed twice with 320 mM trehalose, 20 mM Tris pH 7.3, 1 mM EGTA. Samples were mixed 1:1 with fiducial gold markers (6 or 10 nm gold particles conjugated to protein A, Aurion, Netherlands) and immediately plunge-frozen in liquid ethane on Quantifoil holey carbon grids (Quantifoil Micro Tools, Germany). Single tilt series (typically ± 60°, step size 1.5–2.0°) were collected at 300 kV with an FEI Polara or FEI Titan Krios electron microscope equipped with a post-column Quantum energy filter and a K2 Summit direct electron detector (Gatan) at 6 μm underfocus. Magnifications were chosen to give an object pixel size of 3.5 Å or 3.3 Å, respectively. The total electron dose per tilt series was 90–120 e⁻/Å². If dose fractionation was used, frames were aligned using the IMOD software package (*Kremer et al., 1996*). Tilt series were aligned to gold fiducial markers, binned 2-fold and tomograms were reconstructed by back-projection with IMOD. A final filtering step applying non-linear anisotropic diffusion (*Frangakis and Hegerl, 2001*) was performed to increase contrast. Tomograms were manually segmented with the program Amira (FEI).

## Analysis of cryo-ET data

Distances in tomographic data were analyzed using IMOD. The following parameters were measured for each contact: the radii $r$ of mitochondria perpendicular and parallel to the contact area, the major axis $l_a$ and the minor axis $l_b$ of the contact area assuming elliptical geometry and the distance between the outer membranes $d$. The contact area $A$ was calculated assuming it to be elliptical, $A = (\pi l_a l_b)/4$. The normalized contact area ratio $R$ was calculated, taking the radius r of the smaller of the two involved mitochondria into account, $R = l_a/(2r)$.

## Analysis of *in vitro* mitochondrial fusion reactions by fluorescence microscopy

For fluorescence microscopy analysis, the fusion reactions were fixed with two volumes of 8% formaldehyde in phosphate-buffered saline (PBS). Aliquots were immobilized on microscope slides by mixing 1:1 with 2% low melting point agarose (Sigma-Aldrich) in NMIB. The ratios of fused mitochondria were obtained by dividing the number of GFP and mCherry signals co-localizing with each other over the total number of OM45-GFP (obtained from strain #779) and mito-mCherry (obtained from strain #980) mitochondria (n > 1000). The levels of fused mitochondria were then determined by subtracting the ratios obtained for fusion reactions stopped at the mixing step (at the beginning of the reaction) from those obtained from reactions stopped at t = 45 min (at the end of the reaction).

## Protein extracts and immunoblotting

Cells grown in SD medium were collected during the exponential growth phase ($OD_{600}$ = 0.5–1). Total protein extracts were prepared by the NaOH/trichloroacetic acid (TCA) lysis technique (*Volland et al., 1994*). Proteins were separated by SDS-PAGE 8% and transferred to nitrocellulose membranes (Amersham<sup>TM</sup> Hybond<sup>TM</sup>-ECL; GE Healthcare, UK). Primary antibodies for immunoblotting were mouse anti-Pgk1 (clone 22C5D8; Abcam, UK), rabbit anti-Fzo1 (Covalab, France), anti-Ugo1 (Covalab), anti-Mgm1 (kind gift from Andreas Reichert) and mouse anti-Por1 (clone 16G9E6BC4; Abcam). Primary antibodies were detected by secondary anti-mouse or anti-rabbit antibodies conjugated to horseradish peroxidase (HRP, Sigma-Aldrich), followed by incubation with the Clarity Western ECL kit (Bio-Rad). Immunoblotting images were acquired with a Gel Doc<sup>TM</sup> XR + (Bio-Rad) and processed with the Image Lab 3.0.1 software (Bio-Rad).

## Spot assays

Cultures grown overnight in SD medium were pelleted, resuspended at $OD_{600}$ = 1, and serially diluted (1:10) five times in water. Three microliters of the dilutions were spotted on SD and SG plates and grown for 3 days (dextrose) or 5 days (glycerol) at 30°C.

## Mitochondrial network morphology

Mitochondrial morphology was analyzed in cells expressing mito-GFP from the p426-TEF-mitoGFP plasmid (MC244). Strains were grown in dextrose medium to mid-log phase and fixed with 3.7% formaldehyde. Morphology phenotypes were assessed in at least 100 cells. Data reported are the mean and standard deviation (SD, error bars) of three independent experiments.

## Fluorescence microscopy

Fluorescence microscopy was carried out with a Zeiss Axio Observer Z1 microscope (Carl Zeiss Microscopy GmbH, Germany). For *in vitro* fusion assays a 100X oil immersion objective and the following filter sets were used: 10 Alexa Fluor 489 (Excitation BP 450–490, Beam Splitter FT 510, Emission BP 515–565) for GFP, mCherry (Excitation BP 542–582, Beam Splitter FT 593, Emission BP 604–679) for mCherry. Images were acquired with an SCMOS ORCA FLASH 4.0 charge-coupled device camera (Hamamatsu Photonics K.K., Japan) and the Zen 2012 Package Acquisition/Analyse software before processing with ImageJ. Mitochondrial morphology was analyzed with a 63X oil immersion objective and an FITC filter (Filter set 44, Excitation BP 475/40, Beam Splitter FT 500, Emission BP 530/50) for GFP. Cell contours were visualized with Nomarski optics. Images were acquired with an ORCA-R2 CCD camera (Hamamatsu) and processed with ImageJ.

## Quantum dot labelling

Specific labelling of mitofusins on attached mitochondria was achieved using Quantum Dots coupled to streptavidin (QDot 525ITK streptavidin; Life Technologies; Carlsbad, CA), which required specific biotinylation of Fzo1. For this purpose, the *FZO1* ORF placed under control of its own promoter, was fused at its 3'-end in tandem with a sequence encoding for the Avi tag, a 15 amino-acid peptide that can be recognized and undergo specific biotinylation by the *E. coli* biotin ligase BirA (*Beckett et al., 1999*; *van Werven and Timmers, 2006*). The resulting *pFZO1-AVI* plasmid (MC369) was introduced in *fzo1△* cells using a plasmid shuffling strategy to yield the *FZO1-AVI* yeast strain (MCY1155) that expresses Fzo1-Avi as the sole source of mitofusin in the cell. After verifying that Fzo1-Avi is competent for mitochondrial fusion *in vivo*, mitochondria were purified to promote biotinylation of Fzo1-Avi *in vitro*. Briefly, 0.5 mg of total mitochondrial enriched fractions prepared from FZO1 (MCY1154) or FZO1-Avi (MCY1155) cells were incubated at 25°C for 60 min with 10 µg BirA and 1X biotin in biomix A + B buffer (Biotinylation kit purchased from GeneCopoeia Inc; Rockville, MD) before processing for attachment assays. Following 30 min incubation on ice in Stage 1 buffer, attachment reactions were incubated for 60 min at 4°C with 50 nM streptavidin-coupled QDs before processing for cryo-ET analysis.

## Acknowledgements

We thank Friederike Joos for expert technical assistance in preparing and imaging thin sections and Naïma Belgareh-Touzé for critical reading of the manuscript. This work was funded by the CNRS-INSERM ATIP-Avenir program, the "fondation pour la recherche médicale" (INE 20100518343), a Marie Curie IRG grant (No.276912) and the labex DYNAMO (ANR-11-LABX-0011-DYNAMO) to MC and generous funding of the Max Planck Society to WK. The authors declare no competing financial interests.

## Additional information

### Competing interests

WK: Reviewing editor, *eLife*. The other authors declare that no competing interests exist.

### Funding

| Funder | Grant reference number | Author |
|---|---|---|
| Max-Planck-Gesellschaft | | Werner Kühlbrandt |
| Fondation pour la Recherche Médicale | INE20100518343 | Mickaël M Cohen |
| Marie Curie FP7 | 276912 (Mitofusion) | Mickaël M Cohen |
| Labex DYNAMO | ANR-11-LABX-0011-DYNAMO | Mickaël M Cohen |
| Centre National de la Recherche Scientifique | CNRS-INSERM ATIP-Avenir Program | Mickaël M Cohen |

The funders had no role in study design, data collection and interpretation, or the decision to submit the work for publication.

### Author contributions

TB, Collected EM images, Processed and analyzed the data, Designed the experiments, Wrote the paper, Acquisition of data; LC, Performed *in vitro* attachment/fusion assays as well as cell and molecular biology experiments, Designed the experiments, Analysis and interpretation of data; WK, Wrote the paper; MMC, Designed the experiments, Wrote the paper, Acquisition of data, Analysis and interpretation of data, Contributed unpublished essential data or reagents

### Author ORCIDs

Mickaël M Cohen, http://orcid.org/0000-0002-1372-680X

## Additional files

### Supplementary files

• Supplementary file 1. Table of strains and table of plasmids used in the study.

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
