## [Decision Letter]

Thank you for submitting your article "A mitofusin-dependent docking ring complex triggers mitochondrial fusion" for consideration by *eLife*. Your article has been reviewed by three peer reviewers, one of who is a member of our Board of Reviewing Editors and the evaluation has been overseen by Randy Schekman as the Senior Editor.

The reviewers have discussed the reviews with one another and the Reviewing Editor has drafted this decision to help you prepare a revised submission.

Summary:

In this manuscript, Brandt et al. analyzed structural steps involved in the fusion of the mitochondrial outer membrane by applying electron cryo-tomography to an in vitro mitochondrial fusion assay. The authors centrifuged the mitochondria to attach to each other at 4°C and stimulated fusion by increasing the temperature to 25°C. After centrifugation, the authors observed two types of mitochondrial contacts. The first type of contact, with a relatively short distance of 1-3 nm, includes a regular pattern of globular proteins or protein complexes at the periphery of the contact sites (termed docking ring) based on three-dimensional reconstitution. The second type of contact does not exhibit a docking ring, and globular protein patterns are observed uniformly over the contact surface. The formation of these contact sites requires Fzo1, an outer membrane GTPase that is essential for mitochondrial fusion. Incubation of mitochondria in the presence of GMP-PNP decreased the first type of contact site with a docking ring and increased the second type of contacts, suggesting that GTP hydrolysis is necessary for the formation of contacts with a docking ring from less-organized contacts. To test whether the docking ring contains Fzo1, the authors examined the localization of Fzo1 using the Avi tag and quantum dots and noted that Fzo1 was enriched at the junctions. The authors additionally suggest that fusion initiates at the periphery of contact sites where the docking ring is located (based on one image that they were able to capture). The model in which Fzo1 forms a ring-like pattern at membrane contacts using GTP hydrolysis is interesting. Since structural intermediates of mitochondrial fusion are largely unknown, this study potentially provides new insight into the mechanisms of mitochondrial fusion. However, at the same time, the findings are still preliminary.

Essential revisions:

1) The manuscript presents an interesting model. The formation of a large ring is different from what has been expected. The reviewers appreciate the high technical challenges posed by this problem and the novelty of the approach, but at the same time are skeptical about the model. The main point seems to be that localized tethering is followed by spreading of the contacts to a larger area with Fzo1 moving out to the edges. It is not easy to understand how this sequence of events promotes fusion. The reviewers worry that the spreading could be an artefact of the system; in vitro fusion is slow and relatively inefficient in comparison with fusion in living cells, and thus the single fusion event caught by cryo-EM could be abnormal. Can it be excluded that the large rings may just happen when mitochondria are squished together enough in vitro. It is not clear if this may happen *in vivo*.

In Figure 6, the authors seem to conclude that the site of fusion is adjacent to the docking ring on the basis of one image. The reviewers strongly feel that the single fusion event observed is not enough. Additional images and quantification will be required.

For instance, would lowering the temperature slow down the rate of fusion pore opening and help to obtain more images of this intermediate? In order to capture further fusion intermediates the authors may also consider the possibility of treating docked mitochondria with a crosslinker. As outer membrane fusion is accompanied by a quick disassembly of trans-associated Fzo1 molecules, crosslinked molecules are probably more difficult to disassemble so that local fusion pores might be retained and fusion might not proceed further.

2) In Figure 7, the authors suggest that the overexpression of Fzo1 increases mitochondrial attachment in vitro. However, it is unclear whether these mitochondria are already attached before the assay was performed because Fzo1-overexpressing cells possess aggregated mitochondria. What is the frequency of attached Fzo1-overexpressing mitochondria without centrifugation?

As the authors say, fragmentation caused by overexpression of Fzo1 could result from stalling of the fusion intermediates, because there is too much Fzo1 on the surface. This problem could, however, also result from an imbalance with other proteins, such as Ugo1 or Mgm1, impeding progression through the different stages of fusion.

3) In Figure 8, only limited images are shown in terms of the localization of quantum dots used to localize Fzo1. Based on these images, it is difficult to determine if Fzo1 is truly enriched at the contact sites. Quantification is also critical to support the conclusion. In addition, the localization of other outer-membrane proteins should be included as negative controls.

4) The title should include that the findings are based on an in vitro assay.

The Discussion should also point out the potential limitations due to the in vitro system used.

5) According to the style of *eLife*, the Abstract should mention the biological system used.

[Editors' note: further revisions were requested prior to acceptance, as described below.]

Thank you for resubmitting your work entitled "A mitofusin-dependent docking ring complex triggers mitochondrial fusion in vitro" for further consideration by *eLife*. Your revised article has been favorably evaluated by Randy Schekman (Senior editor) and three reviewers, one of who, Nikolaus Pfanner, is a member of our Board of Reviewing Editors.

While the manuscript is much improved, we include the comments of the three reviewers. You will see that reviewers 2 and 3 suggest two small points. Please include the information about the number of images examined (reviewer 2). You may also consider mentioning the speculation about the presumed sequence of fission events (last paragraph of the comment of reviewer 3).

Reviewer #1:

The authors have addressed the points raised by a substantial amount of additional data and by modification of the text. The main conclusions are now sufficiently supported by the results presented, leading to a very interesting view on mitochondrial fusion.

Reviewer #2:

The manuscript has been much improved and is ready for publication.

The authors evaluated many images to capture fusion intermediates, but did not state the number of images examined. Doing so may help readers appreciate the technical challenge that had to be overcome in this study and give them a sense of the transience of this particular stage.

Reviewer #3:

Brandt et al. included new data to allay concerns raised in the previous reviews. First they show two more instances of fusion using TEM. Second they looked at mitochondrial aggregation when isolated from cells that overexpress fzo1. They detected aggregation even before further incubations. Third, they looked at ugo1 and mgm1 levels in the cells that overexpress fzo1. The levels of ugo1 did not follow those of fzo1, while mgm1 was shifted towards the short form, suggesting that there is an imbalance in the fusion machinery. Lastly, the authors further quantified fzo1 localization with quantum dots.

These are mostly incremental improvements, but I am reassured that fusion along the rims occurs more often, because it was seen two more times with conventional EM. It is still not possible to do statistics on this, and fusion somewhere along the edge of the contact site could result from squishing the mitochondria together, but it does become more believable and it even starts to resemble a reversal of the presumed sequence of fission events in Fawcett's 'The cell' (see Figure 233, Chapter 7). Maybe those earlier images show mitochondrial fusion instead of fission.

---

## [Author Response]

*Essential revisions:*

*1) The manuscript presents an interesting model. The formation of a large ring is different from what has been expected. The reviewers appreciate the high technical challenges posed by this problem and the novelty of the approach, but at the same time are skeptical about the model. The main point seems to be that localized tethering is followed by spreading of the contacts to a larger area with Fzo1 moving out to the edges. It is not easy to understand how this sequence of events promotes fusion. The reviewers worry that the spreading could be an artefact of the system;* in vitro *fusion is slow and relatively inefficient in comparison with fusion in living cells, and thus the single fusion event caught by cryo-EM could be abnormal. Can it be excluded that the large rings may just happen when mitochondria are squished together enough in vitro. It is not clear if this may happen in vivo.*

The *in vitro* mitochondrial fusion assay was set up by the Nunnari group and allowed demonstrating that outer and inner membrane fusion are distinct events (Meeusen *et al.*, 2004). This distinction was manifest by the observation of outer membrane-fused intermediates in which two unfused cristae systems are surrounded by a single outer membrane. By blocking fusion of inner membranes, the occurrence of such outer membrane-fused intermediates was subsequently demonstrated *in vivo* (Chan DC *et al.*, Mol Biol Cell, 2009), confirming that *in vivo* and *in vitro* fusion processes are similar.

The major difference between both systems is that, *in vivo*, mitochondria are brought into proximity by the cytoskeleton. *In vitro*, this essential step is achieved by centrifugation. However, bringing mitochondria into proximity by centrifugation is not sufficient to promote fusion *in vitro*. An additional 10 minute pre-incubation on ice was shown to be required (Meeusen *et al.*, 2004).

Our high-resolution analysis of attachment intermediates reveals the existence of tethered and docked mitochondria and describes the differences between these two states. Most importantly, it establishes that the incubation on ice enables the transition from tethering to docking and demonstrates the essential requirement of GTP hydrolysis, which proves that it is an active process. An important conclusion of the study is that absence of docking systematically correlates with a lower incidence of outer membrane-fused intermediates, and therefore with an inhibition outer membrane fusion. Moreover, mitochondria in the docked configuration are the only intermediates with which we were able to capture fusion events (1 in cryoEM and 2 new events in TEM; see below). Taken together, these observations indicate clearly that docked mitochondria are the initial state in the formation of outer membrane-fused intermediates *in vitro*. Visualization of docking *in vivo* merits further investigation, but this clearly goes well beyond the scope of our study. In the revised manuscript, we extended the Discussion to emphasize the main differences between *in vivo* and *in vitro* mitochondrial fusion (see point 4 below).

*In Figure 6, the authors seem to conclude that the site of fusion is adjacent to the docking ring on the basis of one image. The reviewers strongly feel that the single fusion event observed is not enough. Additional images and quantification will be required.*

For instance, would lowering the temperature slow down the rate of fusion pore opening and help to obtain more images of this intermediate? In order to capture further fusion intermediates the authors may also consider the possibility of treating docked mitochondria with a crosslinker. As outer membrane fusion is accompanied by a quick disassembly of trans-associated Fzo1 molecules, crosslinked molecules are probably more difficult to disassemble so that local fusion pores might be retained and fusion might not proceed further.

We agree that the visualization of additional fused intermediates such as that captured by cryo-EM is desirable. It has to be realized though that the visualization of such inherently rare events by electron cryo-tomography, which samples volumes of a few billionths of a nanoliter at a time, is extremely difficult. We were fortunate to capture one fusion intermediate after incubation of docked intermediates at 4°C and extensive screening. By lowering the temperature we slowed down the rate of fusion pore opening as much as possible, but even so found only one single event. In addition, fusion is, even at 4°C, highly transient, which reduces the capturing probability further. Lowering the temperature to 0°C is unlikely to help, and temperatures below 0°C are obviously excluded in aqueous solution. It might be possible to visualize fusion events by live imaging and fluorescence microscopy, but this would be at much lower resolution and an entirely different study.

While at first sight the suggestion to treat docked mitochondria with crosslinking agents may seem reasonable, it could be (rightly) criticized for producing artefacts, as it carries a high risk of capturing aberrant and non-productive events instead of *bona-fide* docking and fusion intermediates. Moreover, to monitor such experiments by cryo-tomography would be extremely tedious and time-consuming. We conclude that in the context of our present study, crosslinking is not a good approach.

To produce more images of fusion intermediates and to satisfy the reviewer and editor, we resorted to EM analysis of thin sections of mitochondria that were fixed and stained after inducing fusion *in vitro*. In this way, it is possible to sample large populations of attached mitochondria, some of which will have undergone fusion of their outer membranes. While this is still a rare event, we were indeed able to image two more fusion events in this way, in addition to the fusion intermediate captured by cryo-EM. In both cases, the outer membranes are fused near the outer rim of the contact region. These two intermediates are now shown as Figure 6—figure supplement 2 in the revised manuscript. Note that our new EM analysis did not indicate any fusion intermediates other than mitochondria with two *cristae* surrounded by a single outer membrane, as shown in new Figure 6—figure supplement 1. This fully supports our model and indicates that docking results in fusion, which commences near the edge of the contact area, which precedes the apposition and merging of inner membranes occurs. These new results are presented in a paragraph in the revised manuscript.

*2). In Figure 7, the authors suggest that the overexpression of Fzo1 increases mitochondrial attachment* in vitro*. However, it is unclear whether these mitochondria are already attached before the assay was performed because Fzo1-overexpressing cells possess aggregated mitochondria. What is the frequency of attached Fzo1-overexpressing mitochondria without centrifugation?*

This is a good point. To check it, we used fluorescence microscopy to analyze the ratio of attached mitochondria from cells overexpressing Fzo1 as compared to wild-type. This was done before centrifugation, after centrifugation and after 10 minutes incubation on ice (New Figure 7—figure supplement 1). The results indicate that mitochondria from Fzo1-ovexpressing cells are not only more attached before centrifugation but are also more competent for *de novo* attachment *in vitro* (as evidenced by the difference of attachment before and after centrifugation).

*As the authors say, fragmentation caused by overexpression of Fzo1 could result from stalling of the fusion intermediates, because there is too much Fzo1 on the surface. This problem could, however, also result from an imbalance with other proteins, such as Ugo1 or Mgm1, impeding progression through the different stages of fusion.*

In the manuscript, we analyze mitochondrial attachment upon Fzo1 overexpression and conclude that it “induces the formation of protein aggregates that perturb the regulated sequence of events required to reach productive mitochondrial docking.” We did not intend to discuss the molecular details that might explain the fusion deficiency.

We nonetheless fully agree with the reviewers that the fusion defect may be caused by the aggregation of Fzo1 molecules on outer membranes, an imbalance with other proteins such as Ugo1 and Mgm1, or possibly both. This then led us to test the levels of Ugo1 and Mgm1 in whole-cell extracts prepared from wild-type or Fzo1-overexpressing cells with anti-Ugo1 and anti-Mgm1 antibodies (New Figure 7—figure supplement 1). Interestingly, while Ugo1 levels did not vary, which is consistent with a possible imbalance, the ratio between long and short forms of Mgm1 was slightly shifted toward the short form in cells overexpressing Fzo1. This may contribute to the changes in *cristae* morphology upon Fzo1 overexpression as seen in the electron micrographs shown in Figure 7—figure supplement 1.

*3). In Figure 8, only limited images are shown in terms of the localization of quantum dots used to localize Fzo1. Based on these images, it is difficult to determine if Fzo1 is truly enriched at the contact sites. Quantification is also critical to support the conclusion. In addition, the localization of other outer-membrane proteins should be included as negative controls.*

This comment is well taken. It is however important to remember that the enrichment of Fzo1 at mitochondrial contact sites has already been shown by immuno-labelling of TEM sections (Hoppins *et al.*, J Cell Biol 2009). In our study we nevertheless took the challenge of devising a specific method for Fzo1 labelling on cryo-tomograms, partly to confirm this finding, but mainly to demonstrate that mitofusins accumulate in contact regions, where we discovered the docking rings and regular repeats of globular densities. We must also emphasize that *in situ* Q-dot labelling of endogenous proteins in cryo-tomograms has to our knowledge never been published before. The reviewers and editors should be aware that our approach thus represents a technical breakthrough and is very much more challenging than a routine labelling technique such as to immuno-staining of thin plastic sections.

We thank the reviewers for inviting us to show more images of Fzo1 labelling and to quantify our observations, to corroborate our conclusions. The requested changes have been made in the new panels C, D and E of Figure 8. The quantification demonstrates that while Q-Dots were rarely found on the surface of tagged or untagged mitochondria (Figure 8, right graph), they were enriched threefold at the junction of attached FZO1-Avi mitochondria as compared to attached FZO1 mitochondria (Figure 8, left graph). This indicates both that Fzo1 labeling was specific (although weak) and that this specific labeling occurred preferentially in regions where Fzo1 accumulates as shown by Hoppins *et al*. These observations thus confirm that in our tomograms Fzo1 is enriched at mitochondrial contact sites, which is in line with the involvement of this mitofusin in the assembly of docking rings and regular repeats of globular densities.

In light of these results, the necessity to localize other outer-membrane proteins becomes questionable for several reasons. First, the experiment we devised is already controlled for non-specific labelling. Second, it is unlikely that specific labelling for another outer-membrane protein could be achieved unless this factor gets enriched at particular regions of the mitochondrial surface. Third, localizing another protein would require yet another setup to tag the protein, label it with Q-dots and localize it by Cryo-EM, which, taken together, would place an unreasonably heavy burden on us that would take many months.

*4) The title should include that the findings are based on an* in vitro *assay.*

*The Discussion should also point out the potential limitations due to the* in vitro *system used.*

The words “*in vitro*” have been added to the title and the Discussion has been extended to explain and rationalize the main differences between *in vivo* and *in vitro* fusion of outer membranes.

*5) According to the style of eLife, the Abstract should mention the biological system used.*

The Abstract has been edited to emphasize that the *in vitro* attachment and fusion assays were performed with mitochondria isolated from *Saccharomyces cerevisiae* cells.

[Editors' note: further revisions were requested prior to acceptance, as described below.]

*We include the comments of the three reviewers. You will see that reviewers 2 and 3 suggest two small points. Please include the information about the number of images examined (reviewer 2). You may also consider to mention the speculation about the presumed sequence of fission events (last paragraph of the comment of reviewer 3).*

The requested changes have been made as follows: the number of micrographs examined has been added in the Results section and in the Materials and methods; a paragraph that relates to the presumed fission events has been added in the Discussion. We would like to sincerely thank the reviewers and editors for their comments.